# Identifying the lifetime cognitive and socioeconomic antecedents of cognitive state: seven decades of follow-up in a British birth cohort study

M Richards,[1] Sarah-Naomi James,[1] Alison Sizer,[2] Nikhil Sharma,[1,3] Mark Rawle,[1] Daniel H J Davis,[1] Diana Kuh[1]

[1]MRC Unit for Lifelong Health and Ageing at UCL, UCL, London, UK
[2]Epidemiology and Public Health, University College London, London, UK
[3]National Hospital for Neurology and Neurosurgery, University College London Hospitals NHS Foundation Trust, London, UK

**Correspondence to**
M Richards;
m.richards@ucl.ac.uk

## ABSTRACT

**Objectives** The life course determinants of midlife and later life cognitive function have been studied using longitudinal population-based cohort data, but far less is known about whether the pattern of these pathways is similar or distinct for clinically relevant cognitive state. We investigated this for Addenbrooke's Cognitive Examination third edition (ACE-III), used in clinical settings to screen for cognitive impairment and dementia.

**Design** Longitudinal birth cohort study.

**Setting** Residential addresses in England, Wales and Scotland.

**Participants** 1762 community-dwelling men and women of European heritage, enrolled since birth in the Medical Research Council (MRC) National Survey of Health and Development (the British 1946 birth cohort).

**Primary outcome** ACE-III.

**Results** Path modelling estimated direct and indirect associations between apolipoprotein E (*APOE*) status, father's social class, childhood cognition, education, midlife occupational complexity, midlife verbal ability (National Adult Reading Test; NART), and the total ACE-III score. Controlling for sex, there was a direct negative association between *APOE* ε4 and the ACE-III score (β=−0.04 [−0.08 to −0.002], p=0.04), but not between *APOE* ε4 and childhood cognition (β=0.03 [−0.006 to 0.069], p=0.10) or the NART (β=0.0005 [−0.03 to 0.03], p=0.97). The strongest influences on the ACE-III were from childhood cognition (β=0.20 [0.14 to 0.26], p<0.001) and the NART (β=0.35 [0.29 to 0.41], p<0.001); educational attainment and occupational complexity were modestly and independently associated with the ACE-III (β=0.08 [0.03 to 0.14], p=0.002 and β=0.05 [0.01 to 0.10], p=0.02, respectively).

**Conclusions** The ACE-III in the general population shows a pattern of life course antecedents that is similar to neuropsychological measures of cognitive function, and may be used to represent normal cognitive ageing as well as a screen for cognitive impairment and dementia.

## INTRODUCTION

Using the Medical Research Council (MRC) National Survey of Health and Development (NSHD; the British 1946 birth cohort),[1] we

### Strengths and limitations of this study

► The Medical Research Council (MRC) National Survey of Health and Development (NSHD, the British 1946 birth cohort) is a large population-based sample with prospectively obtained information on socioeconomic status and educational attainment, and tested cognitive function from childhood.
► Addenbrooke's Cognitive Examination third edition is an extensive and comprehensive test of cognitive state.
► Path modelling used parameter estimates for incomplete data, thus minimising effects of missing predictor data.
► The path structure of our model may be specific to cohort; NSHD is ethnically homogeneous and experienced selective secondary education and high occupational mobility at labour market entry.
► Replication in more diverse populations is therefore required before our model can be considered generalisable.

demonstrated multiple paths linking four fundamental developmental and social factors to midlife cognitive function: father's socioeconomic position (SEP), childhood cognitive ability, educational attainment and own midlife SEP.[2] To our knowledge such a path model to understand key life course influences on cognitive state, as assessed in clinical practice, has not been undertaken. This is partly because the most frequently used tests, such as the Mini Mental Status Examination (MMSE) and Montreal Cognitive Assessment (MoCA), are brief and have pronounced ceiling effects. It would however be valuable to investigate whether life course paths to cognitive state show a similar pattern as those for other cognitive functions, which would inform theoretical understanding of, and methodology for, studies of cognitive ageing across the full population range. At the most recent NSHD wave at age 69,

Addenbrooke's Cognitive Examination third edition (ACE-III) was administered. This is a more extensive and comprehensive test of cognitive state than the MMSE or MoCA, with a quasinormal distribution. Using this outcome we estimated a path model incorporating childhood SEP, childhood cognitive ability, educational attainment and midlife occupational complexity, and adding two new paths. First, the National Adult Reading Test (NART), an outcome in the original path model, was now included as an intervening variable; we hypothesised that influences on cognitive state operate significantly through this test. Second, the apolipoprotein E (*APOE*) gene was included, the best-known genetic risk factor for dementia; based on previous work,[3] we hypothesised that the ε4 allele of this gene would be negatively associated with the ACE-III score but not with childhood cognition.

## METHODS
### Participants
The NSHD is a representative sample of 5362 males and females born in England, Scotland and Wales in 1 week in March 1946 (http://www.nshd.mrc.ac.uk/nshd). The 24th data collection was conducted between 2014 and 2015 when study members were aged 68–69 years.[1] At age 69, study members still alive and with a known current address in mainland Britain (n=2698) were invited to have a home visit by a trained nurse; 2149 (79.7%) completed a visit and a further 55 (2.0%) completed a postal questionnaire instead. Of the original cohort, 1026 (19.1%) had died, 578 (10.8%) were living abroad, 22 (0.4%) asked for their participation to be restricted to postal contacts, 621 (11.6%) had previously withdrawn from the study and 417 (7.8%) had been lost to follow-up.

### Measures
#### Principal outcome: the ACE -III
The ACE-III is a screen-implemented test of cognitive state, and has been validated as a screening tool for cognitive deficits in Alzheimer's disease and frontotemporal dementia.[4] The ACE-III is divided into five domains: attention and orientation (scored 0–18), verbal fluency (0–14), memory (0–26), language (0–26) and visuospatial function (0–16). Thus, the maximum total score is 100. Due to the inclusion of verbal fluency, the distribution of the total score is quasinormal and avoids the pronounced ceiling effect of most cognitive state tests. A customised version of the ACE-III was administered by iPad using ACEMobile (http://www.acemobile.org/); where this was not possible, a paper version was used. All offline scoring was undertaken by trained personnel. Of the 2149 participants who had a home visit, 32 refused or were unable to undertake the ACE-III. Of the remaining 2117, 35 undertook but did not complete this; and for the remaining 2082, data for 320 were lost through equipment failure. Thus, complete ACE-III data were available for 1762 participants, 82.0% of those who received a home visit.

### Genetic risk
Genetic risk was primarily represented by the *APOE* ε4 allele. Using blood taken at age 53 or 69–71 by a research nurse, KBioscience analysed single nucleotide polymorphism (SNPs) rs429358 and rs7412 to determine *APOE* genotype. Distribution of alleles was as follows (n=2686), ε2/ε2 n=20 (0.74%), ε2/ε3 n=318 (11.84%), ε3/ε3 n=1538 (57.26%), ε2/ε4 n=68 (2.53%), ε3/ε4 n=657 (24.46%) and ε4/ε4 n=85 (3.16%). For analysis, *APOE* genotype was recoded categorically for the presence of ε4 alleles, with carriers of ε2 included as non *APOE* ε4 carriers. Because of difficulties in interpreting potentially opposing effects on cognition, the 68 participants with ε2/ε4 were excluded. Thus, *APOE* was categorised as no ε4 versus heterozygous ε4 or homozygous ε4. For comparison with *APOE*, polygenic scores (PGS) for Alzheimer's disease were calculated for 2768 participants using blood samples taken at age 53 and 60–64. Genotyping was carried out on the NeuroX2 chip. PGSs were created using Allelic Scoring function in PLINK_v1.9. The base data set used to calculate the PGS was the large, two-stage meta-analysis of genome-wide association studies in individuals of European heritage conducted by the International Genomics of Alzheimer's Project (IGAP).[5] Linkage-disequilibrium parameters were set to $r^2 > 0.2$ and a physical distance threshold for clumping SNPs set to 1 Mb. The PGS included the SNPs with a p-value in the IGAP meta-analysis of p<0.05 (n=31 746).

### Early life SEP
Early life SEP was assessed using father's occupational social class and mother's education, which is associated with offspring cognition independently of father's occupation.[6] The former was classified when participants were aged 11 (or at 4 or 15 years if this was unknown) according to the UK Registrar General: professional, managerial, intermediate, skilled manual, semiskilled manual and unskilled; mother's education was coded as primary only versus secondary or any formal qualifications.

### Childhood cognitive function
At 8 years, participants took tests of verbal and non-verbal ability devised by the National Foundation for Educational Research,[7] and administered by teachers or other trained personnel. These tests were as follows: (1) reading comprehension (selecting appropriate words to complete 35 sentences), (2) word reading (ability to pronounce 50 words), (3) vocabulary (ability to explain the meaning of these 50 words) and (4) picture intelligence, consisting of a 60-item non-verbal reasoning test. Scores for each test were standardised to the whole sample, then summed to create a total score representing overall cognitive ability at this age.

### Educational attainment
The highest educational qualification achieved by 43 years was grouped into no qualification, below ordinary

**Table 1** Frequency distributions for APOE group, childhood and midlife SEP, educational attainment, and mean NART and ACE-III scores (for 1762 participants with ACE-III data)

| Variable | n (%) |
|---|---|
| APOE | |
| No ε4 | 1076 (61.1) |
| Heterozygous ε4 | 388 (21.0) |
| Homozygous ε4 | 48 (2.7) |
| Missing | 250 (14.2) |
| Father's social class | |
| Professional | 134 (7.6) |
| Managerial | 372 (21.1) |
| Intermediate | 296 (16.8) |
| Skilled manual | 519 (29.4) |
| Semiskilled | 271 (15.4) |
| Unskilled | 79 (4.5) |
| Missing | 91 (5.2) |
| Mother's education | |
| Primary only | 1151 (65.3) |
| Secondary or any formal qualifications | 421 (23.9) |
| Missing | 611 (10.8) |
| Educational attainment (by age 43) | |
| No qualifications | 403 (22.9) |
| Vocational only | 235 (13.3) |
| Ordinary ('O') level | 340 (19.3) |
| Advanced ('A') level | 497 (28.2) |
| Higher | 2.73 (15.5) |
| Missing | 14 (0.8) |
| NS-SEC occupation (by age 53)* | |
| 1 | 221 (12.5) |
| 2 | 504 (28.6) |
| 3 | 307 (17.4) |
| 4 | 200 (11.4) |
| 5 | 120 (6.8) |
| 6 | 228 (12.9) |
| 7 | 154 (8.7) |
| Missing | 28 (1.6) |
| **Mean (SE)** | |
| NART | 35.6 (0.22) |
| ACE-III | 91.52 (0.14) |

1, higher managerial, administrative and professional occupations; 2, lower managerial, administrative and professional occupations; 3, intermediate occupations; 4, small employers and own account workers; 5, lower supervisory and technical occupations; 6, semiroutine occupations; 7, routine occupations.
ACE-III, Addenbrooke's Cognitive Examination third edition; APOE, apolipoprotein E; NART, National Adult Reading Test; NS-SEC, National Statistics Socio-Economic Classification; SEP, socioeconomic position.

secondary qualifications (vocational), ordinary secondary qualifications ('O' levels and their training equivalents), advanced secondary qualifications ('A' levels and their equivalents) or higher qualifications (degree or equivalent).

### Midlife occupational complexity

Midlife occupational complexity was represented by the National Statistics Socio-Economic Classification (NS-SEC) of the job held at age 53 or earlier if this was missing.[8] This provides a measure of employment relations and the conditions of employment, based on the Standard Occupational Classification: details of individual employment status (ie, employer/employee/self-employed), supervisory position and number of employees in the workplace. For NSHD, information was available on the start and end dates of up to 27 jobs and their NS-SEC categories, which were recoded into seven classes: (1) Higher managerial, administrative and professional occupations. (2) Lower managerial, administrative and professional occupations. (3) Intermediate occupations. (4) Small employers and own account workers. (5) Lower supervisory and technical occupations. (6) Semi-routine occupations. (7) Routine occupations.

### The National Adult Reading Test

The NART assesses ability to pronounce 50 words of increasing difficulty.[9] These words violate conventional pronunciation rules, and are therefore unlikely to be read correctly unless the reader is familiar with them rather than relying on intelligent guesswork. Thus, the NART serves as a measure of general (crystallised) cognitive ability.

All measures were coded so that higher values signified higher status or function.

### Statistical methods

All statistical analyses were conducted using STATA V.15.[10] Path modelling was used to quantify associations between each predictor variable and the ACE-III. Since each of the predictor variables is closely inter-related, the model also quantified their independent interassociations. We hypothesised two key components within this model: (1) strong paths from childhood cognition and the NART to the ACE-III, with modest and weak additional contributions from education and midlife occupational complexity, respectively, and no direct path from childhood SEP[2]; (2) a direct negative path from APOE ε4 to the ACE-III but not via childhood cognition[3] or the NART. No directionality of association was assumed between mother's education and father's social class, or between occupational complexity and the NART; these paths are therefore represented as correlational only. The path model was adjusted for gender, and incorporated full information maximum likelihood (FIML) parameter estimates to include those with item-missingness. FIML is preferable to estimation based on complete data, since FIML estimates tend to be less biased and more reliable than estimates based on list-wise deletion, even when the data deviate from missing at random and are non-ignorable.[11] Three criteria were used to assess model fit: (1) the $\chi^2$ test, although this can be overly

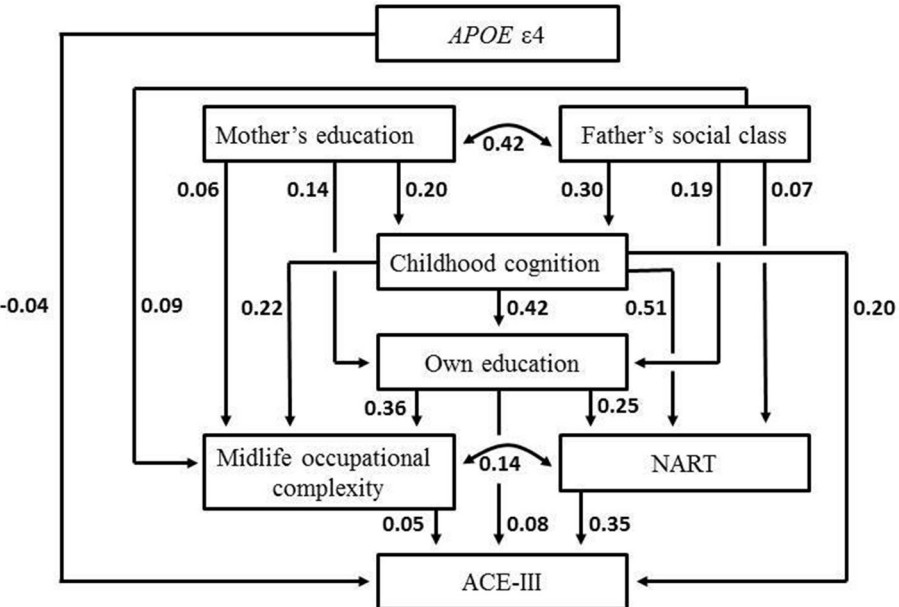

**Figure 1** Path model for the ACE-III total score. ACE-III, Addenbrooke's Cognitive Examination third edition; APOE, apolipoprotein E; NART, National Adult Reading Test.

sensitive to model misspecification when sample sizes are large; (2) the root mean square error of approximation (RMSEA), which gives a measure of the discrepancy in fit per df. It is bounded below by zero, only taking this value if the model fits exactly. If the RMSEA is <0.05, the model is considered a close fit to the data; (3) the comparative fit index (CFI), whose values are restricted to a 0 to 1 continuum, with higher values indicating a better fit. CFI is normally tested against a minimum criterion value of 0.95.

**Patient and public involvement**

Participants have a lifelong association with NSHD. Over the 70 years of the study, the research team has increasingly involved participants, in line with changing norms about conducting cohort studies, starting at age 16 (in 1962) with the annual dissemination of study findings in birthday cards and this continues. Participants have always received a personal letter from the Director whenever they have raised queries or provided additional comments, including suggestions for new topics to study. In the last 10 years, the research team has increased the level of participant involvement through invitations to study events and focus groups to discuss clinical substudies; and a new participant website (www.nshd.mrc.ac.uk/study-members/) was developed in line with their feedback. When piloting new questionnaires and assessments, we recruit patients from General Practitioner (GP) practices or from the University College London Hospital (UCLH) patient and public involvement (PPI) and take into account their feedback when designing the mainstage fieldwork for NSHD participants.

## RESULTS
### Descriptive

As noted, sample size was 1762, the maximum N for the ACE-III. Those who were not interviewed at age 69 for any reason showed no difference in *APOE* ε4 frequency (p=0.72) but had lower childhood cognition and NART scores, and were more likely to be disadvantaged in terms of father's social class, mother's education, own education and occupational complexity (all p<0.001). Those not interviewed were also previously shown to have three or more clinical disorders at the previous assessment (age 60–64), a general health self-rating as poor or fair rather than good, and a longstanding limiting illness, although the latter was not associated with interview participation after controlling for socioeconomic and cognitive characteristics.[1] Of those interviewed at age 69, there were no differences in any of the path variables between those with and without ACE-III data, except for a slight trend for the ACE-III to be missing in those with no educational qualifications ($\chi^2$=9.5, p=0.05). Frequencies for each category of APOE group, childhood and midlife SEP and educational attainment, and means and SDs for the ACE-III and NART, are shown in table 1.

### Path model

Figure 1 shows the path model. All paths are mutually adjusted. Goodness-of-fit statistics indicated that the model was an excellent representation of the data ($\chi^2$=0.15, p=1.0 for analytic vs saturated model; RMSEA=0, p=1.0; CFI=1.0). Gender effects and all non-significant paths (p-value >0.05) are not shown.

The strongest influences on the ACE-III score were from the NART, and from childhood cognition, which was

mainly associated with the ACE-III via educational attainment and the NART, but also directly with the ACE-III. The influence of midlife occupational complexity was more modest, and was itself part-mediated by the NART. There was no direct path from father's social class or mother's education to the ACE-III, but these had independent associations with childhood cognition, educational attainment and midlife occupational complexity, in descending order of magnitude. *APOE* ε4 showed a modest direct negative association with the ACE-III score, but was not associated with childhood cognition or the NART. When the model was rerun replacing *APOE* with the PGS, the path to the ACE-III was of negligible magnitude (β=0.004, 95% CI: −0.031 to 0.038, p=0.82). The paths from the latter to childhood cognition and the ACE-III were also non-significant (β=−0.02, 95% CI: −0.05 to 0.019, p=0.35; β=0.002, 95% CI:−0.04 to 0.04, p=0.9, respectively). However, the path from PGS to NART was significantly negative (β = −0.03, 95% CI: −0.06 to −0.004, p=0.03).

When the model was rerun on the ACE-III subscales, *APOE* ε4 was associated with Attention and Memory, with a similar magnitude to that of the total score (online supplementary table 1); this reached 5% significance when these two scales were combined. Associations between *APOE* ε4 and the Language, Fluency and Visuospatial scales were negligible. Childhood cognition was significantly associated with Attention, Memory and Fluency (negatively in the case of the latter, even though other variables were associated in the expected direction), but not Language or Visuospatial. Education, occupational complexity and the NART were associated with Fluency, Memory and Visuospatial to varying degrees, but none of these were significantly associated with Attention, and only occupational complexity was associated with Language (online supplementary table 1).

## DISCUSSION

In the NSHD, we estimated a path model describing key life course influences on cognitive state using ACE-III. Confirming our main study hypothesis, by far the strongest influence on this outcome was from lifetime cognition, most strongly from general cognitive ability in midlife, assessed by the NART. The NART in turn was particularly influenced by childhood cognition. To a lesser extent, educational attainment was positively associated with the ACE-III, independently of childhood cognition, although the model suggests that this was part-mediated by the NART. Occupational complexity showed more modest effects still, and there were no direct associations between either measure of childhood SEP (mother's education and father's occupational social class) and the ACE-III, although these latter variables were associated with the intervening variables with magnitudes directly proportional to proximity. Finally, there was a direct negative association between the *APOE* ε4 allele and the ACE-III; ε4 was not associated with childhood cognitive function,

or with the NART. The pattern of associations for parental SEP, childhood cognition and education broadly reflect those previously shown in this cohort when the NART was an outcome rather than a predictor,[2] even with an important genetic influence on cognitive function (APOE ε4) controlled. However, it is notable that, with the NART controlled, childhood cognition, education and midlife SEP additionally showed direct associations with the ACE-III, with childhood cognition having the strongest effect, and midlife SEP the weakest.

Major strengths of this study are as follows: (1) the use of a large representative population-based birth cohort; (2) an extensive and comprehensive measure of cognitive state (ACE-III) as an outcome; (3) prospective measures across the life course, including tested childhood cognition, which enabled a comprehensive prospective life course model of mental state; (4) path modelling that uses FIML parameter estimates for incomplete data, thus minimising effects of missing predictor data. Against these strengths, we should note the disproportionate loss to follow-up in those less socially advantaged, with lower prior cognitive function, and with higher physical morbidity. Also, the path structure of our model may be specific to the cohort (NSHD is ethnically homogeneous) and period (NSHD experienced selective secondary education and high occupational mobility at labour market entry). While our previous work suggests a broadly robust path structure in the face of social change,[12] replication in more diverse populations is required before our model can be considered generalisable.

Our path model suggests that cognitive state has a prominent general cognitive ability component, which in turn has cognitive antecedents extending back into childhood. It might be argued that the influence of the NART is a matter of circularity, reflecting the dominance of verbal-based tests within the ACE-III (accounting for 84% of the total score). However, the NART also correlates with non-verbal skills.[9] The most obvious difference between the NART and the ACE-III is that the constituent tests of the latter are 'fluid' measures, sensitive to age and morbidity-associated decline; whereas the former, as a measure of 'crystallised' ability, is stable even in the face of mild dementia.[13] Further follow-up will determine whether the cognitive paths within our model retain their magnitude and pattern as the ACE-III scores change over time. In this context, it is important to note that the present study has not yet incorporated the clinical outcomes of mild cognitive impairment (MCI) and dementia, where a life course approach to the latter has been described.[14] Cognitive decline from approximately the same age has been observed elsewhere when participants with these outcomes were excluded[15]; and also across midlife in NSHD, not explained by concomitant medical conditions,[16] the treatment of which can increase risk of MCI and dementia.[17 18]

In regard to the long-term cognitive antecedents of the ACE-III, the present study extends our previous studies showing that childhood cognition tracks across the life

course even when education, lifetime SEP[2] and adolescent mental health[19] are controlled. This tracking is also consistent with earlier studies in relation to cognitive ageing[20] and risk of dementia[21 22]; and with studies showing that associations between tests of mental state and verbal cognitive ability are strongly explained by childhood cognitive function.[23 24] We also observed an additional direct association between childhood cognition and the ACE-III that was independent of the NART as well as other factors in the model. This is probably because the measures of childhood cognition capture a wider range of function than the NART, including non-verbal reasoning, even though, as noted, the NART itself predicts a comprehensive range of cognitive function.[9]

The next most prominent influence on the ACE-III was from educational attainment, which was primarily based on qualifications through formal education, but also captured qualifications achieved up to early midlife, whether through job training or other paths through adult education. This was associated with the ACE-III even when childhood cognition was controlled. As with childhood cognition itself, the influence of education was largely through the NART, although again there was a modest independent association with the ACE-III, since education also shapes non-verbal cognitive skills.[25] An association between education and subsequent cognition independent of childhood cognition has long been observed[26]; has been replicated in two other birth cohorts[27]; is shown in NSHD to be additive with respect to adult education[28] and responds rapidly to policy.[12] By way of interpretation, it is important to note that education is not just a process of 'cognitive stimulation'. Schools indeed teach specific knowledge, but can also teach practical skills, including how to approach cognitive testing, refine other cognitive skills and shape non-cognitive skills that are likely to have long-term benefit to cognitive function.[19 29] Policies to improve access to education, and widen educational curricula to strengthen all these skills, are likely to have long-term benefits to cognitive ageing, and risk of dementia.

Finally, we should consider the role of *APOE* in the model. This is involved in the transport of cholesterol and other lipids between cellular structures, and ε4 has a higher rate of lipoprotein clearance thus altering its bioavailability.[30] *APOE* is also involved in clearing beta amyloid from the brain, and ε4 may be less efficient at this.[31] A direct association between the ε4 allele and the ACE-III was found in our model; this was of relatively weak magnitude and was not observed with any other variable in the model including prior cognitive function. These findings are consistent with evidence that ε4 zygosity shows a dose–response for Alzheimer's disease[32]; with a study showing no association with childhood cognition although observed in old age in the same cohort[3]; and with parallel evidence from NSHD that decline in verbal memory from age 43 to 69 is faster in *APOE* homozygosity.[33] There is no consensus over whether *APOE* is associated with normal cognitive ageing as opposed to clinical

decline.[3 32–38] However, this may be age-dependent[35]; intriguingly, while no association was found between ε4 and fluid cognitive measures in NSHD at age 53,[33 36] this association is now evident 16 years later, although modestly. In regard to cognitive domain, it is interesting to note the finding that *APOE* ε4 was associated with attention and memory in particular. This is a potentially important finding since three of five neuropsychological tests identified by a meta-analysis as having the highest predictive accuracy for progression from MCI to Alzheimer's disease were of episodic memory.[39] It should also be highlighted that the presence of *APOE* in the model means that the structure and magnitude of the pathways, including those between parental social class and childhood cognition, were independent of this. Adding *APOE* does not of course comprehensively control for genetic influence on cognitive ageing. However, the ε4 allele of this gene is the best-known genetic risk factor for clinically significant cognitive decline.[40]

In contrast to *APOE* ε4, while a PGS for AD was associated with a lower NART score, this was not associated with the ACE-III, although scores based on the same IGAP database used as a reference for the PGS in this study are predictive of AD itself.[41] The lack of association between a PGS for AD and general cognitive ability is consistent with a recent study using the Lothian birth cohort,[42] although these authors found an association for cognitive slope (but not intercept) with a more stringent whole-genome threshold (0.01) than ours (0.05). They suggest that SNPs unrelated to *APOE* ε4 may be overpowering the signal from this; indeed, a systematic analysis of the GenAge database found *APOE* to be one of the top three genes associated with the greatest number of age-related diseases.[43]

In conclusion, the ACE-III in the general ageing population shows a pattern of life course antecedents that is similar to neuropsychological measures of cognitive function. This may not have emerged from studies using briefer tests of cognitive state such as the MMSE, since most of these have ceiling effects outside the clinical context that limit their use as continuous measures. As noted, continuing follow-up of NSHD will elucidate whether the path structure we describe here changes as an increasing number of participants meet clinical criteria for dementia, and the distribution of the ACE-III shifts accordingly.

**Acknowledgements** We thank National Survey of Health and Development (NSHD) study members for their lifelong participation and past and present members of the NSHD study team who helped to collect the data. We also thank Addenbrooke's Cognitive Examination (ACE) Mobile for providing a customised version of the ACE-III for NSHD.

**Contributors** MR conceived the work, conducted the statistical analyses and wrote the manuscript; SNJ advised on the path modelling; SNJ, AS, NS, MR, DHJD and DK contributed to interpretation and writing.

**Funding** This study was supported by the UK Medical Research Council, which provides core funding for the NSHD, and supports the authors by MC_UU_12019/1 (Enhancing NSHD) and /3 (Mental Ageing). DD is additionally funded through a Wellcome Trust Intermediate Clinical Fellowship (WT107467).

SNJ is additionally funded by Alzheimer's Research UK (ARUK-PG2014–1946 and ARUK-PG2017-1946).

**Competing interests**  None declared.

**Patient consent for publication**  Not required.

**Ethics approval**  NRES Queen Square REC (14/LO/1073) and Scotland A REC (14/SS/1009).

**Provenance and peer review**  Not commissioned; externally peer reviewed.

**Data sharing statement**  Bona fide researchers can apply to access the NSHD data via a standard application procedure (further details available at: http://www.nshd.mr.cac.uk/data.aspx).

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
