## [Reviewer comments · BMJ Open]

ARTICLE DETAILS

TITLE (PROVISIONAL)	Identifying the lifetime cognitive and socioeconomic antecedents of cognitive state: seven decades of follow-up in a British birth cohort study.
AUTHORS	Richards, M; James, Sarah-Naomi; Sizer, Alison; Sharma, Nikhil; Rawle, Mark; Davis, Daniel; Kuh, Diana

VERSION 1 - REVIEW

REVIEWER	LAWRENCE WHALLEY UNIVERSITY OF ABERDEEN UNITED KINGDOM
REVIEW RETURNED	15-Jun-2018

GENERAL COMMENTS	Lifetime antecedents of cognitive state: seven decades of follow-up in a national birth cohort study (Richards et al) General impression: This is a follow up paper from a highly regarded group of life course researchers with a substantial record of achievement in understanding sources of individual differences in health and ageing. The analysis reported here continues an approach to their dataset using statistical modelling by path analysis as earlier reported by Richards and Sacker (2003, ref2). Path analysis is based on multiple regression and allows for the analysis of more complicated models that can include “chains” of influence. The method provides a diagram of relationships between outcome and predictor variables. It is similar to the structural equation modelling procedures of comparable data by research groups elsewhere (for example, Chapko et al Age and Ageing 2016; 45: 486–493). The value of the paper under review is limited by lack of clinical data on participants but is strengthened by a new extensive dataset describing cognitive performance at age about 69 and the results of APOE genotyping. The absence of clinical data will disappoint some readers of a general medical journal particularly those interested in determinants of health and well-being in late adulthood. Clinical geriatricians recognise that extracerebral disease can accompany cognitive decline where it may have a causal role. Scientific rationale: The MRC National Survey of Health and Development (the British 1946 birth cohort) longitudinal study is internationally renowned for the comprehensive nature and extent of data systematically collected from birth to late adulthood. Numerous scientific reports from repeated examinations and data-linkage are available from this cohort and have informed public policy (especially on health inequalities) for several decades. Now entering ages at which risk of cognitive decline will rapidly increase, the 1946 birth cohort is uniquely placed to explore gene-social environment interactions in late adulthood and their effects on individual differences in cognitive performance. In their
---

introduction to the cohort when age 69, the authors set the agenda for future longitudinal studies in health and ageing:

“There is growing evidence that biological ageing, manifesting as premature mortality, increased risk of chronic diseases, and decline in physical and cognitive capability with subsequent increases in functional limitations and difficulties with activities of daily living have their origins in environmental exposures and experiences earlier in life [3]. They also share certain underlying ageing processes post maturity that occur at the body system, cellular or molecular levels that lead to reduced physiological reserve [4]. There is also growing evidence that psychological and social wellbeing in later life have their origins earlier in life, albeit based on a somewhat different set of past exposures and experiences than those associated with biological ageing [5, 6]. Less well studied, from a life course perspective, are the lifetime determinants of the common health symptoms and conditions which are often the sequelae of biological ageing, such as chronic pain, incontinence and fatigue, which can impair quality of life and lead to a loss of independence.” (Kuh et al, 2016, ref 1)

This line of reasoning captures exactly the consensus now shared by those gerontologists and geriatricians who recognise that the wealth of experimental data from studies of lower organism support the proposal that intrinsic biological ageing processes make major contributions to risk of age-related disease. This view is emphasised by observations on human progeroid syndromes which demonstrate that disruption of key biological processes can result in the premature onset of multiple age-related pathologies including cognitive impairment. The point arises, however, whether it is useful in 2018 to replicate the original Richards and Sacker (2003) path analysis, though with genetic and better cognitive outcome data, when interest has now shifted toward more complex explorations of interplay between parameters of biological ageing and a wider range of late life deficits including cognitive performance.

There are numerous studies to show that explanatory variables examined here (paternal and participant social class, education, childhood intelligence) are well established correlates of cognitive performance in late adulthood and are shared also with age-related cardiovascular disease, chronic obstructive pulmonary disease, some cancers and all-cause mortality. It is widely accepted among gerontologists that common processes/ mechanisms underpin both ageing and the pathogenesis of multiple age-related diseases and anticipate that targeting common factors in ageing will have a significant benefit for human health.

Specific points:

(a) Sampling: The study sample (N=2698) are survivors of the original birth cohort (N=5362) and comprise 50% of those recruited. The authors limit their account of censoring when reporting censoring only by death or unavailability. Loss to follow-up among this cohort attributable to cognitive impairment is relevant. Severe mental illness (eg Jones et al. Lancet 1994; 344:1398) or disability should be added. To understand other major influences on cognitive function in late adulthood, comorbidities with age-related diseases not limited to neurodegeneration should be included in the sample description. It is possible that the final study sample is not as representative as the original cohort as is implied here. Factors influencing survival to age 69 are relevant and should be addressed; these will include childhood intelligence and APOE status.

(b) Measures: The authors have access to a richly documented extensive database containing much potentially relevant information throughout the life course. The decision to limit their investigation to a small selection of early life variables is based on their aim to extend the original report (Richards and Sacker, 2003) but this is not developed and requires explanation. The principal outcome measure (ACE-III) is a good choice and provides a robust measure of overall mental ability that is normally distributed in this sample. The decision to limit genetic data to the APOE genotype is puzzling. Although it is recognised that APOE is the best established gene with associations consistently replicated in studies of longevity and dementia, the APOE locus is just one of about a dozen from the gerontome that contribute to risk of most age-related diseases (Johnson et al, Aging Cell 2015; 14:809; Fernandes et al, Hum Mol Genet 2016; 25:4804). There is a sufficient range of socioeconomic variables in the 1946 database to explore pathways between networks of molecular genetic data and informative social variables in this cohort and so make a major contribution to the field of cognitive ageing. The choice of so few social variables is disappointing. Early life exposures were limited to paternal socioeconomic position (mothers living alone are not mentioned). The classification of occupations is not referenced and it is unclear if the same classification was used for fathers at participants' birth in 1946 and for offspring in midlife (1999). Educational attainment alone is an insufficient measure of exposure to education as an influence on late life cognition as it fails to capture the range of educational experiences available to this birth cohort especially those opportunities arising after completion of formal schooling. Occupational complexity would add more to processes already known to influence brain ageing and cognitive decline (Suo et al, Neurolmage 2012; 63:1542). The NART may not be as insensitive as stated here "to age and morbidity-associated decline" and it is probably sufficient to omit this remark.

(c) Statistical methods are clearly set out and easy to follow.

These are appropriate to test the aim to extend the original report.

(d) Results: All tables are necessary. More information about cohort participants lost to follow up would be helpful here. It may interest readers unfamiliar with the 1946 cohort to see if childhood cognitive test scores influence attrition from the study and how children impaired by developmental dyslexia fared when tested aged 69. The table summary data would be more informative (and demonstrably representative) if population statistics were included for comparison. The multiple regression is clearly set out.

(e) Figure 1 The path model lies at the heart of this paper and is the most original and novel aspect of their analysis. The obvious interdependence of social and cognitive variables is adequately addressed. The findings are unsurprising and consistent with earlier reports on these associations with late life cognition.

(f) Discussion: Associations between paternal social class, participant social class and social mobility, educational duration and attainments are widely reported strong predictors of adult mental health, morbidity and mortality in late life. To this limited extent, these results are foreseeable and are confirmatory.

However, the inclusion of APOE and ACE-III data in the path model is innovative and adds weight to these established associations. Adjustments to analysis using childhood cognition are widely reported by the Edinburgh and Aberdeen groups and are a strength of this and other similar studies. It is tricky, therefore, to accept the claim (p14, lines 48-49) that this is "...the

	first comprehensive prospective life course model of mental state” for two reasons. First, that the model is not comprehensive and, second, comparable limited life course studies are reported not only from the Edinburgh and Aberdeen groups but also from longitudinal studies elsewhere where early life cognitive data were retained and molecular genetic data are now available (for example, de Vries et al, Neurobiology of Aging. 2017, 55:91).
--	--

REVIEWER	Antony Bayer Cardiff University, Wales, UK
REVIEW RETURNED	17-Jun-2018

GENERAL COMMENTS	An interesting question with results clearly presented and strengths and weaknesses discussed, including the issue of circularity. Statistical analysis appears to be appropriate although I am not a statistician. Some minor issues:  1. Can the authors provide a reference to support their statement in the Introduction that ACE-11 is "the most extensive and comprehensive test of cognitive state available"? 2. The second sentence of the Results refers to those "without ACE-111 data". Can the authors clarify who they mean - those receiving home visit? those invited? or ?? 3. In Table 1, NART and ACE-111 scores must be errors. Have they been swapped around by mistake? Also, does NART refer to number of errors or number correct?
--

REVIEWER	Stephen Aichele University of Geneva, Switzerland
REVIEW RETURNED	19-Jun-2018

GENERAL COMMENTS	This study is clearly presented and based on a compelling data set. However, I believe the authors have done themselves and the data a disservice by taking an overly narrow focus on a singular outcome (ACE-III total score) – especially as they describe this measure as the “most comprehensive test of cognitive state available.” In brief, the authors considered the following predictors of ACE-III performance: SES (father’s occupational class; midlife SES), childhood cognition (primarily verbal ability), education, midlife verbal ability (NART), and APOE e4. Not surprisingly, they found (a) that childhood cognition most strongly predicted total score on ACE-III, an association partially mediated by NART performance, and (b) that APOE e4 homozygosity was negatively linked to total ACE-III score but not to childhood cognition or NART performance. Outcome (a) certainly seems like a circular effect (i.e., verbal ability predicting verbal ability). The authors have anticipated this criticism in their discussion of the outcomes, but I’m still not convinced this result adds anything meaningful to the existing literature (e.g., it essentially shows that childhood IQ predicts general cognitive performance in later life, which, as noted by the authors, has been shown by others in multiple prior studies).
---

	So then, what stands out? The authors make the case that the ACE-III is a “clinically relevant” measure, whereas prior such studies used outcomes that, presumably, were not clinically relevant. Yet no information is provided as to just what “relevance” means here; e.g., in terms of cutoff scores or percentages of study participants either “at risk” or “diagnostic” for dementia. If this is the key point differentiating this study from others, a more forceful case must be made in the introduction and more nuanced discussion of the implications is necessary. I would suspect that outcome (b) links APOE e4 to ACE-III performance mainly due to the memory component of the ACE-III (whereas the NART and childhood IQ measures did not directly assess memory). It does not appear that the authors considered this possibility, which seems odd given the abundance of studies linking APOE e4 to memory impairment, including a prior study using this same sample cited by the authors. Overall, this research would be more compelling (and informative) had the authors examined performance on subtests of the ACE-III, in addition to the total score, as outcomes. This should be trivial to implement methodologically and not require much additional reporting space. It would, however, require more nuanced discussion of the results. At the least a follow-up analysis to assess the connection between APOE e4 and scores on ACE-III subtests (especially memory) could be reported as supplemental materials. Minor points:  1) Abstract: please report sample size(s) and effect size estimates (standardized regression weights would be OK). 2) The authors tout the ACE-III as being “the most comprehensive test of cognitive state” several times. This seems a bold statement and requires some justification, linked to appropriate references. 3) How was the NART mediation effect estimated? If I recall, MPLUS makes it possible to test for this without resorting explicitly to model comparisons.
--	---

VERSION 1 – AUTHOR RESPONSE

Reviewer 1 (Lawrence Whalley)

General impression:

“The point arises ... whether it is useful in 2018 to replicate the original Richards and Sacker (2003) path analysis, though with genetic and better cognitive outcome data, when interest has now shifted toward more complex explorations of interplay between parameters of biological ageing and a wider range of late life deficits including cognitive performance.”

RESPONSE: We are in general agreement with this point, but would argue here that not enough is known about the life course determinants of cognitive state, as measured by instruments routinely used for clinical diagnostic purposes. In this context we feel that our study makes a genuine original contribution to the understanding of these instruments.

“The value of the paper under review is limited by lack of clinical data on participants but is strengthened by a new extensive dataset describing cognitive performance at age about 69 and the results of APOE genotyping. The absence of clinical data will disappoint some readers of a general medical journal particularly those interested in determinants of health and well-being in late adulthood. Clinical geriatricians recognise that extracerebral disease can accompany cognitive decline where it may have a causal role.”

RESPONSE: If the reviewer is primarily referring to neurodegenerative diseases, at age 69 when the outcome for our study (ACE-III) was measured, the 1946 birth cohort was still relatively young for clinical outcomes, most importantly dementia. Eventually we will be able to re-estimate our path model when these outcomes are more common (e.g. when in the age 75-80 age bracket). In the meantime, we consider that identifying key lifetime influences on the ACE-III is a valuable step.

Sampling:

“Loss to follow-up among this cohort attributable to cognitive impairment is relevant. Severe mental illness (eg Jones et al. Lancet 1994; 344:1398) or disability should be added. To understand other major influences on cognitive function in late adulthood, comorbidities with age-related diseases not limited to neurodegeneration should be included in the sample description. It is possible that the final study sample is not as representative as the original cohort as is implied here. Factors influencing survival to age 69 are relevant and should be addressed; these will include childhood intelligence and APOE status.”

RESPONSE: We have now added comparative details of those lost to follow-up vs. interviewed at age 69, including morbidity (page 11, Results, Descriptive) as well as the path variables; and a comment on this is added to the Discussion under limitations (page 16, paragraph 2; also relevant to point (d) of this reviewer). More generally, while we agree that childhood cognition and APOE status are predictors of survival (indeed, have shown this for childhood cognition in separate NSHD publications), we hope that use of full information maximum likelihood parameter estimates for item-missingness minimized bias from this source.

Measures:

“The decision to limit genetic data to the APOE genotype is puzzling. Although it is recognised that APOE is the best established gene with associations consistently replicated in studies of longevity and dementia, the APOE locus is just one of about a dozen from the gerontome that contribute to risk of most age-related diseases...”

RESPONSE: we have now documented the comparison of results after substituting APOE ϵ 4 for a polygenic score for Alzheimer’s disease. Rather than strengthen the path to ACE-III this association was of negligible magnitude. We therefore feel that APOE ϵ 4 is a better genetic predictor, and indeed this is implied by the second reference provided by the reviewer (Fernandes et al.).

“There is a sufficient range of socioeconomic variables in the 1946 database to explore pathways between networks of molecular genetic data and informative social variables in this cohort and so make a major contribution to the field of cognitive ageing. The choice of so few social variables is disappointing. Early life exposures were limited to paternal socioeconomic position (mothers living alone are not mentioned).”

RESPONSE: There were few mothers living alone in the early years of the 1946 birth cohort; however, we added mother’s education, which is shown here and elsewhere to influence cognitive development independently of father’s social class.

“Educational attainment alone is an insufficient measure of exposure to education as an influence on late life cognition as it fails to capture the range of educational experiences available to this birth cohort especially those opportunities arising after completion of formal schooling.”

RESPONSE: We have now expanded the original variable, which represented level of educational qualification attained by age 26, to include qualifications attained by age 43, whether by job training or other routes through adult education.

“Occupational complexity would add more to processes already known to influence brain ageing and cognitive decline...”

RESPONSE: We have replaced RG social class at or by age 53 with the more detailed equivalent of the National Statistics Socio-Economic Classification (NS-SEC).

While none of these changes in measures substantively altered the model results, we agree that they significantly improve the representation of the chosen life course factors.

“The NART may not be as insensitive as stated here “to age and morbidity-associated decline” and it is probably sufficient to omit this remark.”

RESPONSE: We have duly removed this.

“Statistical methods are clearly set out and easy to follow. These are appropriate to test the aim to extend the original report.”

RESPONSE: Thank you.

“More information about cohort participants lost to follow up would be helpful here. It may interest readers unfamiliar with the 1946 cohort to see if childhood cognitive test scores influence attrition from the study and how children impaired by developmental dyslexia fared when tested aged 69. The table summary data would be more informative (and demonstrably representative) if population statistics were included for comparison.”

RESPONSE: As noted above, we have now added information on characteristics of those lost to follow-up. The long-term consequences of dyslexia, including cognitive function, is the topic of a separate study currently under development. Comparison with British population statistics is included in a previous NSHD paper (Stafford et al. Eur J Ageing 2013;10:145–157), and given the above essential revisions we feel that inclusion of this here would unduly burden the paper.

Discussion

“It is tricky, therefore, to accept the claim (p14, lines 48-49) that this is “...the first comprehensive prospective life course model of mental state”...”

RESPONSE: While we would still argue that ours is a comprehensive prospective life course model of cognitive state, we agree that it is not the first, and have amended this statement accordingly. We acknowledge that there are numerous studies of cognitive ageing that incorporate measures of early SEP, education and adult occupation as covariates, including the de Vries et al. study cited by the reviewer; however, we have also amended the above sentence to refer to our study as a path model, which we maintain is an under-investigated approach.

Reviewer 2 (Anthony Bayer)

“An interesting question with results clearly presented and strengths and weaknesses discussed, including the issue of circularity. Statistical analysis appears to be appropriate although I am not a statistician.”

RESPONSE: Thank you.

“Can the authors provide a reference to support their statement in the Introduction that ACE-11 is “the most extensive and comprehensive test of cognitive state available”?”

RESPONSE: The question of which is the most extensive and comprehensive test of cognitive state depends on the criteria for selecting these; for example the CAMCOG assessment battery is more extensive, whereas we are referring to single instruments widely used in clinical setting. We have therefore toned down this statement, and now simply refer to the ACE-III as an (rather than the most) extensive and comprehensive test.

“The second sentence of the Results refers to those “without ACE-111 data”. Can the authors clarify who they mean - those receiving home visit? those invited? or ??

RESPONSE: Reasons for missing ACE-III data are documented in the Methods section (page 7 paragraph 1).

“In Table 1, NART and ACE-111 scores must be errors. Have they been swapped around by mistake? Also, does NART refer to number of errors or number correct?”

RESPONSE: A statement has now been added that all measures are coded so that higher value corresponds to higher status or better performance (page 10, top).

Reviewer 3 (Stephen Aichele)

“This study is clearly presented and based on a compelling data set.”

RESPONSE: Thank you.

“The authors make the case that the ACE-III is a “clinically relevant” measure, whereas prior such studies used outcomes that, presumably, were not clinically relevant. Yet no information is provided as to just what “relevance” means here; e.g., in terms of cutoff scores or percentages of study participants either “at risk” or “diagnostic” for dementia.”

RESPONSE: We have added the statement that the ACE-III “has been validated as a screening tool for cognitive deficits in Alzheimer’s disease and frontotemporal dementia” (page 6 paragraph 2).

“I believe the authors have done themselves and the data a disservice by taking an overly narrow focus on a singular outcome (ACE-III total score)...” “I would suspect that outcome (b) links APOE e4 to ACE-III performance mainly due to the memory component of the ACE-III (whereas the NART and childhood IQ measures did not directly assess memory). It does not appear that the authors considered this possibility, which seems odd given the abundance of studies linking APOE e4 to memory impairment, including a prior study using this same sample cited by the authors. “ “Overall, this research would be more compelling (and informative) had the authors examined performance on subtests of the ACE-III, in addition to the total score, as outcomes.”

RESPONSE: We have followed this reviewer’s recommendation, and investigated the path model using the subscales of the ACE-III as outcomes as well as the total score, which is now referred to in the text and the data shown in a Supplementary table added. The reviewer is indeed correct that the APOE-ACE-III path is mainly due to the memory scale (and also the attention scale), which we feel has important implications for progression to clinical outcomes.

“Abstract: please report sample size(s) and effect size estimates (standardized regression weights would be OK).”

RESPONSE: We have added sample size for the ACE-III and effect sizes to the Abstract along with 95% confidence intervals and p values, as requested by the editor.

“The authors tout the ACE-III as being “the most comprehensive test of cognitive state” several times. This seems a bold statement and requires some justification, linked to appropriate references.”

RESPONSE: Following the response to Reviewer 1 to this point, the statement about the ACE-III being the most comprehensive test of cognitive state has been modified.

“How was the NART mediation effect estimated?”

RESPONSE: We did not explicitly test mediation effects, so the statement about the path between childhood cognition and the ACE-III being part-mediated by the NART is now removed.

Beyond these responses to the reviewer comments, we should also point out four further changes:

1. We discovered an error in the previous manuscript in regard to APOE $\epsilon 4$. Our results should have referred to any $\epsilon 4$ (heterozygotes or homozygotes) compared to no $\epsilon 4$, rather than homozygotes compared to heterozygotes or no $\epsilon 4$. This is now amended.
2. Due to a serious and unexplained fault that developed with Mplus, we conducted all new analyses in STATA version 15.
3. In the interest of keeping down the length of the manuscript following the revisions, we have removed the initial regression tests (text and Table 2). On reflection we feel that this does not add anything substantive to the study; path modelling is of course based on regression in any case, and its principal advantage is to allow a view of internal paths within the same model.
4. Dr. Mai Stafford has now left the Unit, and has withdrawn from authorship of this and all other ongoing Unit publications. We are indebted to Ms. Alison Sizer for deriving the NSSEC variable at age 53 as part of her PhD thesis, and have therefore added her as an author.

We thank the reviewers for their extremely helpful comments, and hope that our revisions are an adequate response. We believe that the changes have led to a better paper, and hope that you will now consider this favourably for publication in BMJ Open.

VERSION 2 – REVIEW

REVIEWER	Lawrence Whalley University of Aberdeen, UK
REVIEW RETURNED	07-Oct-2018

GENERAL COMMENTS	I agree with the authors' changes to the original manuscript. Notwithstanding, I remain doubtful about the lack of clinical data. Extracerebral disease is not neurodegenerative disease. At this stage, it may be sufficient to cite work showing that at about age 70, at a general population level, cognitive decline is not explained by concomitant medical conditions (e.g. Psychol Aging 31: 166–175) but their treatments (BMJ 2018;361:k1315) are highly relevant. RESPONSE: If the reviewer is primarily referring to neurodegenerative diseases, at age 69 when the outcome for our study (ACE-III) was measured, the 1946 birth cohort was still
---

	relatively young for clinical outcomes, most importantly dementia. Eventually we will be able to re-estimate our path model when these outcomes are more common (e.g. when in the age 75-80 age bracket). In the meantime, we consider that identifying key lifetime influences on the ACE-III is a valuable step.
--	--

VERSION 2 – AUTHOR RESPONSE

Review 1 comment: I agree with the authors' changes to the original manuscript. Notwithstanding, I remain doubtful about the lack of clinical data. Extracerebral disease is not neurodegenerative disease. At this stage, it may be sufficient to cite work showing that at about age 70, at a general population level, cognitive decline is not explained by concomitant medical conditions (e.g. Psychol Aging 31: 166–175) but their treatments (BMJ 2018;361:k1315) are highly relevant.

RESPONSE: We thank Professor Whalley for this further advice, and have amended our manuscript accordingly, also taking this opportunity to refer to his lancet Neurology article on a life course approach to late-onset dementia. With regard to iatrogenic cognitive effects of medication, in addition to Richardson et al. suggested by Professor Whalley, we have also cited a relevant NSHD study on effects of polypharmacy (new reference 17). Thus our additional text is as follows:

“In this context it is important to note that the present study has not yet incorporated the clinical outcomes of Mild Cognitive Impairment (MCI) and dementia, where a life course approach to the latter has been described[14]. Cognitive decline from approximately the same age has been observed elsewhere when participants with these outcomes were excluded[15]; and also across midlife in NSHD, not explained by concomitant medical conditions[16], the treatment of which can increase risk of MCI and dementia[17,18].”

We thank Professor Whalley, and hope that this further revision is an adequate response.